# RTop-K: Ultra-Fast Row-Wise Top-K Selection for Neural Network Acceleration on GPUs

[*]**Xi Xie**
University of Connecticut
xi.xie@uconn.edu

[*]**Yuebo Luo**
University of Minnesota - Twin Cities
luo00466@umn.edu

[*]**Hongwu Peng**
University of Connecticut
hongwu.peng@uconn.edu

[+]**Caiwen Ding**
University of Minnesota - Twin Cities
dingc@umn.edu

## Abstract

Top-k selection algorithms are fundamental in a wide range of applications, including high-performance computing, information retrieval, big data processing, and neural network model training. In this paper, we present RTop-K, a highly efficient parallel row-wise top-k selection algorithm specifically designed for GPUs. RTop-K leverages a binary search-based approach to optimize row-wise top-k selection, providing a scalable and accelerated solution. We conduct a detailed analysis of early stopping in our algorithm, showing that it effectively maintains the testing accuracy of neural network models while substantially improving performance. Our GPU implementation of RTop-K demonstrates superior performance over state-of-the-art row-wise top-k GPU implementations, achieving an average speed-up of up to $11.49\times$ with early stopping and $7.29\times$ without early stopping. Moreover, RTop-K accelerates the overall training workflow of MaxK-GNNs, delivering speed-ups ranging from 11.97% to 33.29% across different models and datasets.

The GPU implementation can be found on Github[†].

## 1 Introduction

Top-k selection is a classic algorithmic challenge that involves identifying the k largest or smallest elements from n input elements based on some predefined ranking criteria. The top-k selection algorithm has been widely applied in many traditional scenarios, such as high-performance computing (HPC) (Muneer, 2021), information retrieval (IR) (Ding & Suel, 2011), big data (Gaihre et al., 2019), and data mining (Malkov & Yashunin, 2018). Today, the top-k algorithm is increasingly applied in the training and inference of neural network models. For example, the Avg-TopK (Özdemir, 2023) pooling method has achieved more successful results in image classification accuracy and transfer learning models compared to traditional methods. TopK-SGD (Shi et al., 2019) applied to gradient sparsification techniques significantly reduces the communication traffic without obvious impact on the model accuracy. Combining top-k with sparse training (Jayakumar et al., 2021) can maintain constant sparsity and perform well while reducing resource requirements. In a study (Cui et al., 2021), a top-k attention loss function was introduced to address the top-k ranking prediction problem.

Graph Neural Networks (GNNs) have drawn tremendous attention in the past years due to their unique ability to extract latent information from graph data (Hu et al., 2020). In the design and acceleration of GNN training and inference, GPU platforms have become the prevalent choice due to their multiple advantages. Firstly, compared to other processing hardware, GPUs provide superior processing power and memory throughput (Li et al., 2018). For example, the NVIDIA A6000 GPU boasts an impressive computation capability of 38.7 Tera FLOPS and a memory throughput 768

---

[*]These authors contributed equally. [+]Corresponding author.

[†]https://github.com/xiexi51/RTopK

GB/s. Secondly, many leading supercomputers (such as Aurora and Eagle (LiveScience, 2023)) use GPUs as their primary computing resource. Thirdly, many applications and services related to deep learning and neural networks are developed and deployed on GPU platforms. However, GNN training and inference still typically pose strict challenges on latency and throughput (Xie et al., 2023).

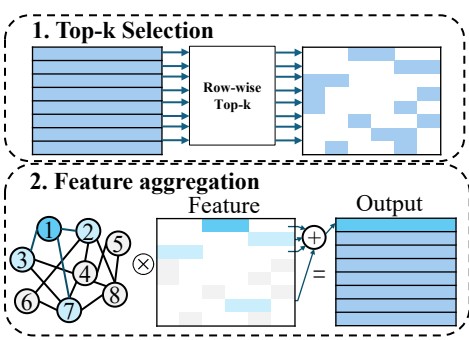

Figure 1: The core operation of MaxK-GNN, which introduces row-wise top-k selection into the GNN workflow to provide non-linearity and acceleration.

Recently, MaxK-GNN (Peng et al., 2024) has achieved great success in the acceleration and optimization of GNN training and inference on GPUs. As shown in Figure 1, this work introduces row-wise top-k selection before the feature aggregation step in GNNs, which not only provides nonlinearity in GNNs to optimize the model's expressive ability but also demonstrates that performing SPMM operations in GNNs with the row-wise top-k-processed right hand matrix can achieve several times speedup over traditional workflows while maintaining good model accuracy. The top-k selection operation in Max-GNN necessitates performing a large-scale row-wise top-k computation, i.e., executing top-k operations simultaneously across a batch of vectors on GPUs.

Traditional top-k algorithms and their GPU implementations (Gaihre et al., 2021; Zhang et al., 2023; Li et al., 2024) are typically optimized for single queries or limited batched queries, that is, for a large vector or a small batch of large vectors (typically with a batch size not exceeding 100). However, the optimization focus for traditional scenarios differs from the row-wise top-k algorithm required for GNN training and inference. Implementing and optimizing row-wise top-k algorithms on GPUs pose challenges in terms of dispersion, parallelism, and efficiency. Since row-wise top-k involves performing top-k operations on a large batch of vectors simultaneously, and each vector's length corresponds to the hidden dimension of the neural network layer (which usually does not exceed 1024), it is crucial to allocate only a small and appropriate amount of GPU resources for each vector. Under these limited resource constraints, the various optimization methods proposed for large vectors in traditional top-k implementations may be overly complex and inefficient. We should seek simple and efficient algorithms tailored to this scenario.

Additionally, we must consider the requirements and characteristics of applying row-wise top-k in neural networks. We only need to select the values of the top-k elements in each row and their indices in the vector. We do not need to perform sorting at all; neither the k selected elements in each row nor the remaining elements require sorting. Furthermore, given the neural network's tolerant and robust nature, we can explore the feasibility of approximate top-k to further accelerate the overall algorithm.

To efficiently implement row-wise top-k on GPUs for neural network applications, we introduce RTop-K, a highly efficient parallel top-k selection algorithm designed for a large batch of limited-size vectors, with the capability of approximation to further enhance the speed of the row-wise top-k algorithm without compromising the accuracy of the neural network model.

We summarize our contributions as follows:

- We provide a comprehensive summary of GPU top-k selection algorithms and analyze the performance limitations of state-of-the-art GPU implementations in the row-wise top-k selection scenario.

- We propose a binary search-based top-k selection algorithm and provide a theoretical analysis of the effects of early stopping.

- We implement the binary search-based top-k selection algorithm on the GPU and conduct comprehensive tests, demonstrating that it outperforms state-of-the-art row-wise top-k GPU implementations, with early stopping having minimal impact on testing accuracy.

## 2 PRELIMINARY AND RELATED WORKS

### 2.1 TOP-K ALGORITHMS

The heap-based top-k algorithm (Cormen et al., 2009) uses a min-heap to maintain the top-k elements by comparing and replacing the heap root when a larger element is encountered. QuickSelect (Dashti et al., 2013) leverages a partition-based approach similar to quicksort to find the k-th largest element. The bucket-based algorithm (Yang et al., 2024) divides data into buckets, sorting only relevant ones to find the top-k elements, which is effective for uniformly distributed data. RadixSelect (Alabi et al., 2012) uses digit-wise sorting to identify the top-k elements efficiently. The bitonic top-k algorithm (Shanbhag et al., 2018) employs bitonic sorting to merge and find the top-k elements in parallel.

When considering these algorithms, we must take into account their suitability for GPU implementation and the optimization requirements for specific problem scenarios. For example, the heap-based top-k algorithm is not well-suited for parallelization on GPUs because it relies on complex tree structure operations and element-wise comparisons and swaps. Although QuickSelect, RadixSelect, and the bitonic top-k algorithm can be successfully implemented on GPUs, they still require considerable data access and resource demands when operating on a vector. This makes it difficult to optimize for row-wise top-k scenarios, where a large batch of limited size vectors requires top-k selection simultaneously, necessitating simplified operations and limited resource usage per vector. The bucket-based top-k algorithm is more friendly to row-wise top-k scenarios but still requires further simplification to enhance performance.

### 2.2 GPU ARCHITECTURE

The architecture of NVIDIA GPUs consists of an array of multithreaded Streaming Multiprocessors (SMs) designed to execute thousands of threads concurrently. A function that runs on a GPU is called a kernel.

**Thread and Memory Hierarchy.** NVIDIA GPUs organize threads into warps, with each warp containing 32 threads that execute the same instruction simultaneously. Warps are grouped into blocks, which reside on the same Streaming Multiprocessor (SM) and can communicate via shared memory, a fast on-chip memory space. Blocks are further grouped into grids for specific kernel launches. Threads access data from multiple memory spaces: device memory (large but slower, accessible by all threads), shared memory (low-latency, for communication within a block), and registers (fastest, partitioned among threads on an SM). The usage of registers can affect the number of blocks that can be active on an SM.

**Warp-Level Primitives**. Warp-level primitives are a set of operations that allow threads within a warp to cooperate and communicate efficiently. These include:

- **Synchronization primitive**: Ensures that all threads reach the same point in execution before proceeding.

- **Shuffle primitive**: Allows threads to exchange values within a warp.

- **Ballot primitive**: Enables threads to collectively determine which threads meet a specified condition by generating a mask representing the threads that satisfy the condition.

- **Counting primitive**: Counts the number of set bits in a given mask, often used in conjunction with the ballot primitive.

The flexible use of warp-level primitives is crucial for designing high-performance kernels, as the efficiency of information sharing through these primitives can even surpass that of using shared memory.

### 2.3 GPU TOP-K IMPLEMENTATIONS

Dr. Top-k(Gaihre et al., 2021) is a delegate-centric system that helps reduce the workload of GPU top-k computations, including Radix Select, Bucket Select, and Bitonic Select. It achieves this by

dividing the input into sub-ranges and selecting delegates from them, along with performing multi-GPU optimizations. A work(Zhang et al., 2023) proposed two optimization methods, AIR Top-k and GridSelect. AIR Top-k employs an iteration-fused design and adaptive strategy to minimize CPU-GPU communication and memory traffic, while GridSelect uses a shared queue and parallel two-step insertion to decrease costly operations, enhancing parallel top-k efficiency on GPUs. A recent RadixSelect implementation RadiK(Li et al., 2024) utilizes an optimization framework tailored for high memory bandwidth and resource utilization, along with an adaptive scaling technique for enhanced robustness, that supports larger k values with high efficiency.

However, the above state-of-the-art GPU implementations are optimized for limited batches of large vectors. For instance, Dr. Top-k, AIR Top-k, and RadiK are designed for scenarios where the vector size is on the order of $2^{20}$ (about one million elements), and the batch size does not exceed 100. This is not suitable for row-wise top-k applications, where the typical vector size is less than 1024, and the batch size can reach millions.

PyTorch's top-k implementation (Pytorch, 2024) is suitable for row-wise top-k operations. It uses RadixSelect as the underlying method, which, as analyzed in Section 2.1, is overly complex for each limited-size vector. Moreover, its selection results are sorted, which is also unnecessary. Although it can handle large batch sizes, its efficiency is suboptimal in scenarios where a minimalistic top-k selection is critical for each single vector.

---

**Algorithm 1** Binary Search-based Top-k Selection Algorithm

---

**Require:** Vector $\mathbf{v}$ of size $M$, integer $k$
**Ensure:** Top-$k$ largest elements and their indices in $\mathbf{v}$
1: $min \leftarrow \min(\mathbf{v})$
2: $max \leftarrow \max(\mathbf{v})$
3: $\epsilon \leftarrow \epsilon' \cdot max$
4: $cnt \leftarrow 0$
5: **while** $max - min > \epsilon$ **do**
6:    $thres \leftarrow \frac{min+max}{2}$
7:    $cnt \leftarrow |\{i \mid \mathbf{v}_i \geq thres\}|$
8:    **if** $cnt < k$ **then**
9:      $max \leftarrow thres$
10:   **else if** $cnt > k$ **then**
11:      $min \leftarrow thres$
12:   **else**
13:      **break**
14:   **end if**
15: **end while**
16: **if** $cnt < k$ **then**
17:   $\mathbf{elems}, \mathbf{indices} \leftarrow \{(\mathbf{v}_i, i) \mid \mathbf{v}_i \geq thres\}$
18:   Append the first $k - cnt$ pairs of $\{(\mathbf{v}_i, i) \mid min \leq \mathbf{v}_i < thres\}$ to $\mathbf{elems}, \mathbf{indices}$
19: **else**
20:   $\mathbf{elems}, \mathbf{indices} \leftarrow$ the first $k$ pairs of $\{(\mathbf{v}_i, i) \mid \mathbf{v}_i \geq thres\}$
21: **end if**
22: **return** $\mathbf{elems}, \mathbf{indices}$

---

## 3   RTop-K Framework

The row-wise top-k operation involves finding the largest (or smallest) $k$ elements and their indices in each row of a matrix. Without loss of generality, we assume finding the largest $k$ elements. Suppose a matrix has $N$ rows and $M$ columns; the problem is equivalent to performing top-k selection on $N$ vectors of length $M$ simultaneously. Since $N$ can be extremely large and $M$ is limited, we need to apply a simplified algorithm to each vector, ensuring that the algorithm can execute quickly with very limited computational resources and memory access. We adopt a binary search-based top-k algorithm, which is even more convenient to execute than the bucket top-k algorithm.

### 3.1   Binary Search-based Top-k Selection Algorithm

The algorithm, as illustrated in Fig. 2, first retrieves the $min$ and $max$ values of the vector, and then uses several iterations of binary search to determine a threshold $thres$. The $min$, $max$, and $thres$ values are updated in each iteration, and the loop terminates when the number of elements filtered by the current threshold equals $k$. One corner case that must be considered is when $\mathbf{v}$ contains multiple equal or very close elements, and these elements happen to be near the "borderline" during top-k selection. In this scenario, relying solely on $thres$ for selection makes it difficult or even impossible to extract exactly $k$ elements. It can be verified that in such cases, these borderline elements are always located between $min$ and $thres$. Therefore, a second filtering step using $min$ is sufficient to supplement the selection and ensure that exactly $k$ elements are chosen.

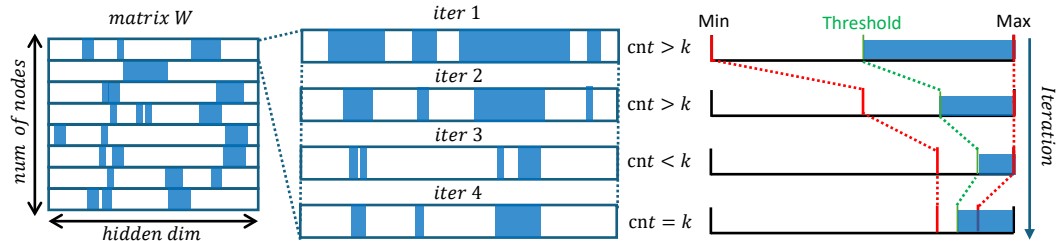

Figure 2: Illustration of the binary search-based top-k selection algorithm.

Table 1: Cumulative percentage of iterations where the loop exits for different $k$ values ($\epsilon = 10^{-4}, M = 256$). Results are based on $10^5$ repeated experiments for each $k$.

| Iteration | k=16 | k=32 | k=64 | k=96 | k=128 |
|---|---|---|---|---|---|
| 3 | 4.13% | 2.71% | 1.96% | 1.34% | 1.58% |
| 4 | 8.98% | 5.32% | 3.52% | 3.00% | 2.81% |
| 5 | 17.90% | 10.64% | 6.92% | 5.84% | 5.59% |
| 6 | 33.86% | 21.40% | 13.87% | 11.72% | 11.15% |
| 7 | 54.43% | 38.84% | 27.12% | 23.29% | 22.11% |
| 8 | 72.38% | 59.17% | 46.64% | 41.35% | 39.93% |
| 9 | 84.53% | 76.00% | 66.21% | 61.48% | 60.35% |
| 10 | 91.88% | 86.81% | 80.68% | 77.37% | 76.62% |
| 11 | 95.81% | 93.03% | 89.79% | 87.64% | 87.18% |
| 12 | 97.89% | 96.45% | 94.70% | 93.57% | 93.31% |
| 13 | 98.97% | 98.21% | 97.35% | 96.70% | 96.60% |
| 14 | 99.52% | 99.12% | 98.67% | 98.34% | 98.25% |
| 15 | 99.76% | 99.53% | 99.34% | 99.20% | 99.17% |
| 16 | 100.00% | 100.00% | 100.00% | 100.00% | 100.00% |
| Average Exit | 7.60 | 8.29 | 8.95 | 9.52 | 9.60 |

The parameter $\epsilon$ represents the precision of the algorithm, which determines the maximum width of the borderline and imposes an upper bound on the number of iterations in the loop. When $\epsilon$ is set to 0 (which is effectively equivalent to the minimal distinguishable precision of the float type), Algorithm 1 becomes an exact top-$k$ selection algorithm.

---

**Algorithm 2** Binary Search-based Top-k Selection Algorithm with Early Stopping

---

**Require:** Vector $\mathbf{v}$ of size $M$, integer $k$, integer $max\_iter$
**Ensure:** Top-$k$ largest elements and their indices in $\mathbf{v}$
1: $min \leftarrow \min(\mathbf{v})$
2: $max \leftarrow \max(\mathbf{v})$
3: **for** $iter \leftarrow 1$ to $max\_iter$ **do**
4:     $thres \leftarrow \frac{min+max}{2}$
5:     $cnt \leftarrow |\{i \mid \mathbf{v}_i \geq thres\}|$
6:     **if** $cnt < k$ **then**
7:        $max \leftarrow thres$
8:     **else**
9:        $min \leftarrow thres$
10:     **end if**
11: **end for**
12: $\mathbf{elems}, \mathbf{indices} \leftarrow \{(\mathbf{v}_i, i) \mid \mathbf{v}_i \geq min\}$
13: **return** first $k$ pairs of $\mathbf{elems}, \mathbf{indices}$

---

Table 1 presents the statistical results of the iteration counts at which the algorithm exits for different values of $k$, with the vector's size $M = 256$. For each $k$ value, $10^5$ repeated experiments were conducted, with the vector initialized with normally distributed elements. It can be observed that the average iteration at exit ranges from 7.6 to 9.6, and the probability of the iteration count being less than or equal to 13 exceeds 95%.

Algorithm 1 summarizes the complete binary search-based top-k algorithm process. It contains a number of branching conditions, and the loop length executed by each warp can be different. We attempt to further simplify it.

Given the inherent robustness of neural networks, we can explore the feasibility of incorporating early stopping into the algorithm. We present the pseudocode for the early stopping algorithm and then conduct a numerical analysis.

As shown in Algorithm 2, the introduction of early stopping further simplifies the algorithm, with the main loop forcefully exiting in no more than $max\_iter$ iterations. The collection phase uses $min$ as the threshold, ensuring that only one-

Table 2: Statistics of early stop top-k selection for Different $k$ Values and Maximum Iterations (M=256). $E_1(\%)$ represents the average relative error between the maximum element in early stop top-k selection and the maximum element in the optimal top-k selection. $E_2(\%)$ represents the average relative error between the minimum element in early stop top-k selection and the minimum element in the optimal top-k selection. Hit(%) represents the overlap ratio between the early stop top-k selection and the optimal top-k selection.

| | $k = 16$ | | | $k = 32$ | | | $k = 64$ | | | $k = 96$ | | | $k = 128$ | | |
|---|---|---|---|---|---|---|---|---|---|---|---|---|---|---|---|
| Iter | $E_1(\%)$ | $E_2(\%)$ | Hit(%) | $E_1(\%)$ | $E_2(\%)$ | Hit(%) | $E_1(\%)$ | $E_2(\%)$ | Hit(%) | $E_1(\%)$ | $E_2(\%)$ | Hit(%) | $E_1(\%)$ | $E_2(\%)$ | Hit(%) |
| 2 | 12.6 | 20.17 | 45.85 | 13.46 | 30.68 | 37.81 | 7.12 | 25.03 | 51.78 | 4.42 | 17.80 | 69.59 | 4.6 | 24.73 | 70.93 |
| 3 | 8.01 | 13.13 | 54.29 | 6.22 | 13.19 | 60.32 | 4.44 | 12.40 | 69.04 | 3.39 | 12.94 | 74.41 | 2.78 | 13.23 | 79.33 |
| 4 | 4.93 | 7.64 | 68.35 | 3.47 | 7.05 | 74.46 | 2.47 | 6.55 | 80.51 | 1.99 | 6.82 | 84.33 | 1.6 | 7.24 | 87.34 |
| 5 | 3.52 | 5.20 | 77.36 | 2.20 | 4.31 | 83.19 | 1.47 | 3.70 | 87.88 | 1.18 | 3.91 | 90.49 | 0.97 | 4.29 | 92.34 |
| 6 | 2.90 | 4.33 | 81.57 | 1.62 | 3.17 | 87.62 | 0.99 | 2.39 | 91.83 | 0.77 | 2.57 | 93.77 | 0.62 | 2.90 | 95.03 |
| 7 | 2.67 | 4.10 | 83.17 | 1.38 | 2.79 | 89.51 | 0.79 | 1.87 | 93.68 | 0.61 | 2.00 | 95.33 | 0.47 | 2.30 | 96.35 |
| 8 | 2.61 | 4.06 | 83.68 | 1.31 | 2.69 | 90.19 | 0.71 | 1.72 | 94.35 | 0.55 | 1.82 | 95.94 | 0.41 | 2.11 | 96.86 |

pass collection is needed. Table 2 summarizes the hit rate (overlap ratio) and the average relative error between the early stopping top-k selection with different $max\_iter$ settings and the optimal top-k selection. The experiments were conducted with vectors of size $M = 256$ consisting of normally distributed elements, and $10^5$ repeated experiments for each condition. When $max\_iter \geq 5$ for $k \geq 32$ ($max\_iter \geq 6$ for $k = 16$), both the maximum element and the minimum element in the early stopping top-k selection have an average relative error of no more than 5%. For $k \geq 64$, only 4 iterations are needed for the hit rate between the early stopping top-k selection and the optimal top-k selection to exceed 80%. These results indicate that the early stopping top-k selection is numerically stable and controllable. We will further test the impact of early stopping top-k selection on model accuracy in the experimental section.

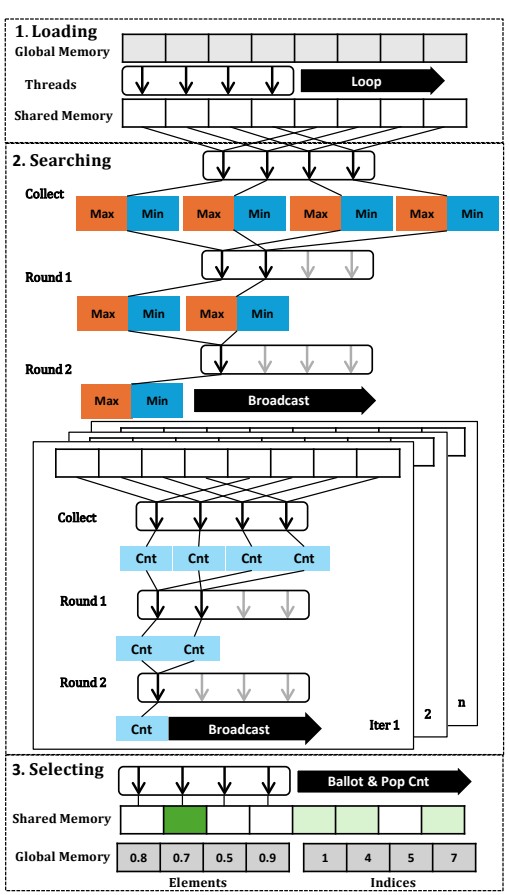

Figure 3: GPU implementation of the binary search-based top-k selection algorithm.

## 3.2 GPU IMPLEMENTATION DESIGN

Both Algorithm 1 and Algorithm 2 are well-suited for GPU implementation, where a single warp processes a single vector of size $M$. Fig. 3 illustrates the GPU implementation design, which can be divided into three stages: loading, searching, and selecting.

**Loading stage:** In this stage, each vector is loaded from global memory into the corresponding shared memory, which is done by a warp looping through it. A synchronization barrier is set at the end of this stage.

**Searching stage:** In this stage, each vector is handled by a single warp, assuming the warp contains $w$ threads (Fig. 3 illustrates $w = 4$, while in actual hardware environments $w = 32$). The key point of the implementation is that the $max$, $min$, and $thres$ values at the beginning and in each iteration need to be synchronized across all threads within the warp. This can be achieved using a combination of classic tree-reduction and broadcast primitives. The first step is to obtain the $max$ and $min$ of the vector. The vector is divided into $\lceil M/w \rceil$ parts, with each thread responsible for extracting the maximum and minimum elements within its assigned part. Then, a tree-reduction using the shuffle primitive is performed in $\log_2 w$ steps

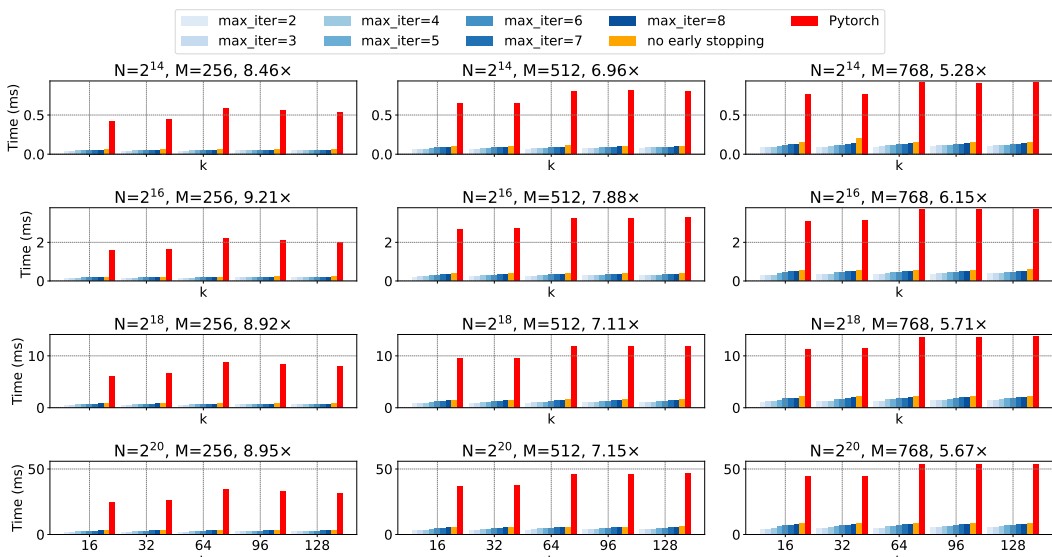

Figure 4: Comparison of kernel execution time (ms) between RTop-K with different early stopping $max\_iter$ values and without early stopping ($\epsilon = 10^{-16}$), against PyTorch for various configurations of $(N, M, k)$, where $N = 2^{14}, 2^{16}, 2^{18}, 2^{20}$, $M = 256, 512, 768$, and $k = 16, 32, 64, 96, 128$. The average speedup of the no early stopping version for each $(N, M)$ setting is indicated in the title of each subplot.

to obtain the maximum and minimum elements across the entire warp, and these values are broadcasted to all threads within the warp. The second step is to perform a binary search to determine the selection threshold. In each iteration, the count of elements exceeding the current threshold is accumulated and broadcasted using the same tree-reduction method. Then, each thread uses $cnt$ to simultaneously update the $max$, $min$, and $thres$ values. Once the exit condition is met or the maximum number of iterations is reached, the final threshold is obtained.

**Selecting Stage:** A single warp performs a one-pass or two-pass selection over each vector. Let $cnt$ be the number of elements greater than or equal to the final threshold $thres$. The selection applies two conditions separately: selecting elements where $x \geq thres$, and, if needed, selecting additional elements where $thres > x \geq min$.

If $cnt \geq k$, the first condition is sufficient to produce $k$ elements, and the second condition is skipped. Otherwise, all elements satisfying the first condition are selected, and the remaining $k - cnt$ elements are supplemented using the second condition.

To implement this efficiently, we use the ballot primitive to merge and broadcast the selection masks across threads within a warp. The popcnt primitive is then used to compute the inclusive prefix sum of selected elements for each thread. Based on the prefix sum, each thread finalizes its selection decision: if the prefix sum exceeds $k$, its selection is disabled. The values and indices of the final selected elements are written to the output buffer in global memory.

This design requires no data writes outside of registers, except for loading the vector and dumping the results. During the searching and selecting stages, warp-level primitives are utilized to achieve highly optimized inter-thread collaboration. Moreover, each warp operates independently in parallel, maintaining high overall efficiency.

## 4 EXPERIMENTS

### 4.1 SETUP AND CONFIGURATION

The CUDA source code of RTop-K is compiled using NVCC, version 12.6, and executed on an NVIDIA A6000 platform running Ubuntu 22.04. We conduct comprehensive performance tests on the RTop-K kernel, covering various input matrix dimensions, with the number of rows $N$ ranging

Table 3: Average speed up of RTop-K compared to PyTorch implementation ($\epsilon = 10^{-16}$ for No Early Stopping) across different $M$ values.

| Max Iteration | 2 | 3 | 4 | 5 | 6 | 7 | 8 | No Early Stopping |
|---|---|---|---|---|---|---|---|---|
| $M$=256 | 13.07 | 12.32 | 11.46 | 10.86 | 10.32 | 9.88 | 9.55 | 8.88 |
| $M$=512 | 11.66 | 11.37 | 10.43 | 9.51 | 8.87 | 8.34 | 7.98 | 7.27 |
| $M$=768 | 9.73 | 9.44 | 8.72 | 7.75 | 7.16 | 6.78 | 6.46 | 5.72 |
| Average | 11.49 | 11.04 | 10.20 | 9.37 | 8.79 | 8.34 | 7.99 | 7.29 |

Table 4: Graph data and the baseline testing accuracy of the MaxK-GNN based GNN model along with the percentage of time spent on row-wise top-k operations during training.

| GNN Model | | GraphSAGE | | GCN | | GIN | |
|---|---|---|---|---|---|---|---|
| Graph | #Nodes | Acc(%) | Top-k Prop(%) | Acc(%) | Top-k Prop(%) | Acc(%) | Top-k Prop(%) |
| Ogbn-products | 2449029 | 80.08 | 19.81 | 76.6 | 19.61 | 77.77 | 19.67 |
| Yelp | 716847 | 61.09 | 26.09 | 48.26 | 25.84 | 43.16 | 25.92 |
| Reddit | 232965 | 96.74 | 11.66 | 95.18 | 11.61 | 94.96 | 11.62 |
| Flickr | 89250 | 53.44 | 26.86 | 50.42 | 26.78 | 51.73 | 26.73 |

from $2^{14}$ to $2^{20}$, hidden dimensions $M$ ranging from 256 to 768, and $k$ values ranging from 16 to 128. In all cases, we evaluate the speed of RTop-K with different early stopping settings, including $max\_iter$ values from 2 to 8, as well as no early stopping ($\epsilon = 10^{-16}$). The results are compared against the row-wise top-$k$ implementation provided by PyTorch (Pytorch, 2024), which is the state-of-the-art row-wise top-$k$ implementation supporting a large number of rows. The latency measurements are conducted using the Nsight Compute tool (NVIDIA, 2023).

We also integrate the RTop-K kernel into MaxK-GNN models to evaluate the overall speedup of the entire training process and the impact of different early stopping settings on test accuracy. The evaluation covers three models in MaxK-GNN: GraphSAGE (Hamilton et al., 2017), GCN (Kipf & Welling, 2016), and GIN (Xu et al., 2019). The graph datasets used include Flickr (McAuley & Leskovec, 2012), Yelp (Zeng et al., 2020), Reddit (William L. Hamilton, 2017), and Ogbn-products (Hu et al., 2020).

## 4.2 RTOP-K KERNEL EVALUATION

Fig. 4 presents a comprehensive time profiling of RTop-K compared to the PyTorch implementation. It can be observed that RTop-K demonstrates significant speed improvements over PyTorch across various early stopping $max\_iter$ settings. Even with no early stopping, RTop-K significantly outperforms PyTorch in all scenarios. Moreover, the dimension that has the most impact on the speed-up ratio is $M$, while $N$ and $k$ have relatively smaller effects. Table 3 summarizes the average speed-up of RTop-K relative to PyTorch for different values of $M$. When $M = 256$, the speed-up varies from $9.55\times$ to $13.07\times$ across different $max\_iter$ settings, and even with no early stopping, the speed-up is still as high as $8.88\times$. On average, the speed-up ranges from $7.99\times$ to $11.49\times$ with early stopping, and $7.29\times$ with no early stopping.

We observe that the speed-up with no early stopping is close to that of $max\_iter = 8$, which indicates that although the binary search requires many iterations in the worst-case, it typically exits early in most cases. This observation is consistent with the results presented in Table 1. Even with a few bad cases, the overall kernel speed remains unaffected.

## 4.3 MODEL TRAINING AND TESTING PERFORMANCE EVALUATION

Table 4 summarizes the proportion of time spent on row-wise top-k operations in several MaxK-GNN training instances. It is evident that row-wise top-k operations account for a substantial portion of the training time, ranging from 11.61% in the Reddit GCN training to 26.86% in the Flickr GraphSAGE training. This indicates that optimizing row-wise top-k operations is meaningful for improving their training efficiency.

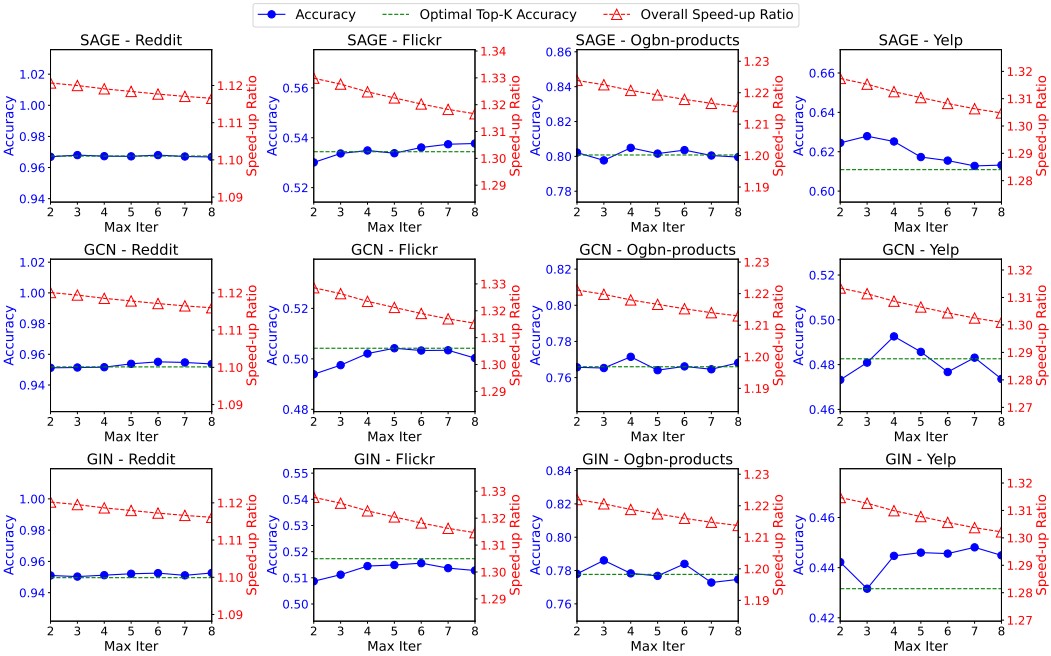

Figure 5: Overall training speed-up ratio and testing accuracy of applying RTop-K to various MaxK-GNN model training processes on different graphs. Setting: $N = \#$Nodes, $M = 256$, $k = 32$.

The impact of applying RTop-K on speed and accuracy with different early stopping settings in the actual training of these models is shown in Fig. 5, with the setting of $N = \#$Nodes, $M = 256$, and $k = 32$. For all GNN models and all graphs, the application of RTop-K effectively accelerates the overall training workflow. Specifically, under different $max\_iter$ settings, the average overall training speed-up for Reddit ranges from 11.97% to 12.21%, for Flickr from 32.48% to 33.29%, for Ogbn-products from 22.00% to 22.74%, and for Yelp from 31.21% to 32.42%.

It can be observed that the testing accuracy of the models remains high. Except for the GIN training on Flickr, the testing accuracy across different $max\_iter$ settings for other cases fluctuates around the testing accuracy achieved with the optimal row-wise top-k selection. In many cases, applying early stopping for row-wise top-k selection even results in higher testing accuracy. This is also a manifestation of the inherent robustness of GNNs.

## 5  CONCLUSION

In this paper, we presented RTop-K, a highly efficient parallel row-wise top-k selection algorithm for GPUs. By employing a binary search-based approach, RTop-K significantly accelerates top-k operations while maintaining the accuracy of neural network models, as confirmed by our theoretical analysis. Comprehensive kernel evaluations demonstrate that RTop-K outperforms state-of-the-art GPU implementations, achieving an average speed-up of up to $11.49\times$ with early stopping and $7.29\times$ without early stopping. The evaluation of the overall MaxK-GNN training workflow with RTop-K shows that RTop-K provides overall speed-ups ranging from 11.97% to 33.29% across different models and datasets, with early stopping having almost no impact on testing accuracy.

## 6  ACKNOWLEDGMENTS

This research was supported in part by NSF SHF-2505770, Semiconductor Research Corporation (SRC) Artificial Intelligence Hardware program.

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

## A  THE EXPECTATION OF THE ITERATION COUNTS OF ALGORITHM 1

Assume that a vector $\mathbf{v}$ of length $M$ has elements that follow a normal distribution $N(\mu, \sigma^2)$. The probability that an element $x$ exceeds a threshold $thres$ is given by $P(x > thres) = 1 - \Phi\left(\frac{thres - \mu}{\sigma}\right)$, where $\Phi$ is the cumulative distribution function of the normal distribution. If the expected number of elements selected from $\mathbf{v}$ is $k$, the expectation of $thres$, denoted as $E(thres)$, satisfies:

$$M \cdot \left(1 - \Phi\left(\frac{E(thres) - \mu}{\sigma}\right)\right) = k \implies E(thres) = \mu + \sigma \cdot \Phi^{-1}\left(1 - \frac{k}{M}\right) \tag{1}$$

Considering the distinguishable interval $\delta$ between the $k$-th and $(k+1)$-th largest elements, the length of this interval is:

$$\delta = \frac{1}{M \cdot f(E(thres))} \tag{2}$$

where

$$f(E(thres)) = \frac{1}{\sigma\sqrt{2\pi}} \exp\left(-\frac{(E(thres) - \mu)^2}{2\sigma^2}\right)$$

is the probability density at $E(thres)$. The length of the initial search interval $D$ is given by:

$$D = \max(\mathbf{v}) - \min(\mathbf{v}) \approx 2\sigma\sqrt{2\ln M} \tag{3}$$

Each iteration of binary search halves the search interval length and moves closer to $E(thres)$. The expected number of iterations $E(n)$ required for the algorithm to exit is determined by the search interval shrinking to within $\delta$. Thus, $E(n)$ can be approximated as:

$$E(n) \approx \log_2\left(\frac{D}{\delta}\right) = \log_2\left(2\sigma\sqrt{2\ln M} \cdot M \cdot f(E(thres))\right)$$

$$= \log_2\left(2M\sqrt{\frac{\ln M}{\pi}}\right) - \frac{1}{2\ln 2}\left(\Phi^{-1}\left(1 - \frac{k}{M}\right)\right)^2 \tag{4}$$

We compared the calculation results of Equation (4) with more detailed experimental results, as shown in Table 5. It can be observed that the results match well, but $E(n)$ is always slightly larger than the measured average exit. This could be because the estimation of the initial search interval, $D \approx 2\sigma\sqrt{2\ln M}$, is valid only when $M$ is sufficiently large. When $M$ is not large enough, the lack of tail samples causes the actual initial search interval to be smaller.

## B  A COMPREHENSIVE ANALYSIS OF THE PERFORMANCE OF RTOP-K WHEN APPLIED TO VARYING VECTOR SIZES

Our design fixes one warp to process one vector. As the vector size $M$ increases, the shared memory required per warp also increases. Given that the available shared memory per block has a limit, on the A6000 GPU, we allocate only $\lfloor 8192/M \rfloor$ warps per block. For $M > 8192$, our current shared memory-based acceleration strategy cannot be directly applied.

For $M \leq 8192$, the speedup of RTop-K relative to PyTorch is shown in Figure 6.

Considering the lower-bound speed of RTop-K (no early stopping version):

- When $M$ is below 1280, RTop-K achieves a $4.9\times$ to $12.5\times$ speedup over PyTorch.
- When $M$ is between 1280 and 3072, RTop-K achieves a $2.3\times$ to $4.9\times$ speedup over Py-Torch.
- When $M$ is between 3072 and 6144, RTop-K achieves a $1.1\times$ to $2.3\times$ speedup over Py-Torch.
- When $M$ is between 6144 and 8192, RTop-K is slower than PyTorch, with only the early stopping version using a small $max\_iter$ still being faster than PyTorch.

Theoretically, as shown in Equation (4), the expected number of search iterations for Algorithm 1 is:

Table 5: Cumulative percentage of iterations where the loop exits in Algorithm 1, for different $M, k$ values, with $\epsilon = 0$. Experimental results are based on $10^4$ repeated experiments for each $M, k$ couple, and theoretical values $E(n)$ are also provided.

| Iters | M, k | | | | | | | | | | | | | |
|---|---|---|---|---|---|---|---|---|---|---|---|---|---|---|
| | 256 64 | 256 128 | 1024 64 | 1024 128 | 1024 256 | 1024 512 | 4096 64 | 4096 128 | 4096 256 | 4096 512 | 8192 64 | 8192 128 | 8192 256 | 8192 512 |
| 1 | 0.12 | 1.5 | 0 | 0 | 0.02 | 0.53 | 0 | 0 | 0 | 0 | 0 | 0 | 0 | 0 |
| 2 | 0.17 | 1.5 | 1.25 | 0.17 | 0.02 | 0.53 | 0.41 | 0.41 | 0.18 | 0 | 0.12 | 0.23 | 0.24 | 0.02 |
| 3 | 1.98 | 1.62 | 1.27 | 0.5 | 0.62 | 0.54 | 0.43 | 0.41 | 0.2 | 0.08 | 0.3 | 0.24 | 0.25 | 0.02 |
| 4 | 3.48 | 2.9 | 1.93 | 1.31 | 0.99 | 0.7 | 1.54 | 0.63 | 0.36 | 0.33 | 1.11 | 0.7 | 0.32 | 0.15 |
| 5 | 6.88 | 6.06 | 4.05 | 2.42 | 1.67 | 1.25 | 2.83 | 1.34 | 0.76 | 0.61 | 2.43 | 1.3 | 0.7 | 0.31 |
| 6 | 13.96 | 11.64 | 8.18 | 4.49 | 3.24 | 2.64 | 5.4 | 2.84 | 1.64 | 1.06 | 4.68 | 2.62 | 1.46 | 0.75 |
| 7 | 27.3 | 23.03 | 16.09 | 9.42 | 6.41 | 5.16 | 11.13 | 5.99 | 3.37 | 2.15 | 9.18 | 5.28 | 2.98 | 1.6 |
| 8 | 46.47 | 40.18 | 30.63 | 18.77 | 12.7 | 10.19 | 21.56 | 12.05 | 6.89 | 4.15 | 19.25 | 10.62 | 5.59 | 3.27 |
| 9 | 66.34 | 60.57 | 50.71 | 35.21 | 24.91 | 19.69 | 39.04 | 24.37 | 13.69 | 8.23 | 34.8 | 21.5 | 11.49 | 6.72 |
| 10 | 81.07 | 76.76 | 69.08 | 55.43 | 43.92 | 35.98 | 60.23 | 42.59 | 26.65 | 16.71 | 55.41 | 38.13 | 22.42 | 13.3 |
| 11 | 90.11 | 87.67 | 82.99 | 72.93 | 63.57 | 55.7 | 76.59 | 62.55 | 46.7 | 31.29 | 73.09 | 58.45 | 39.96 | 25.94 |
| 12 | 95.17 | 93.48 | 90.71 | 84.79 | 78.45 | 73.32 | 87.42 | 78.11 | 66.36 | 52.28 | 85.21 | 75.11 | 60.66 | 44.97 |
| 13 | 97.46 | 96.8 | 95.19 | 91.9 | 88.37 | 85.06 | 93.42 | 87.9 | 81.06 | 70.87 | 92.16 | 86.37 | 77.12 | 64.65 |
| 14 | 98.68 | 98.35 | 97.59 | 95.86 | 93.82 | 92.47 | 96.48 | 93.18 | 89.91 | 83.91 | 96.29 | 92.63 | 87.55 | 80.25 |
| 15 | 99.46 | 99.21 | 98.79 | 97.95 | 96.87 | 96.32 | 98.24 | 96.3 | 94.73 | 91.34 | 98.15 | 96.39 | 93.48 | 89.3 |
| 16 | 99.73 | 99.56 | 99.43 | 98.89 | 98.46 | 98.06 | 99.21 | 98.14 | 97.35 | 95.39 | 99.11 | 98.19 | 96.73 | 94.46 |
| 17 | 99.85 | 99.72 | 99.71 | 99.51 | 99.18 | 99.03 | 99.59 | 99.02 | 98.51 | 97.65 | 99.48 | 99.18 | 98.35 | 97.29 |
| 18 | 99.96 | 99.85 | 99.9 | 99.73 | 99.62 | 99.44 | 99.77 | 99.45 | 99.2 | 98.8 | 99.77 | 99.59 | 99.21 | 98.66 |
| 19 | 99.99 | 99.93 | 99.95 | 99.84 | 99.84 | 99.67 | 99.87 | 99.75 | 99.64 | 99.38 | 99.89 | 99.76 | 99.54 | 99.33 |
| 20 | 99.99 | 99.98 | 99.97 | 99.9 | 99.93 | 99.85 | 99.95 | 99.87 | 99.84 | 99.76 | 99.98 | 99.89 | 99.8 | 99.69 |
| 21 | 99.99 | 99.99 | 99.98 | 99.95 | 99.97 | 99.93 | 99.97 | 99.94 | 99.95 | 99.95 | 99.99 | 99.98 | 99.94 | 99.84 |
| 22 | 99.99 | 99.99 | 99.99 | 99.96 | 99.99 | 99.95 | 99.99 | 99.98 | 99.98 | 99.97 | 99.99 | 99.98 | 99.96 | 99.9 |
| 23 | 99.99 | 99.99 | 99.99 | 99.99 | 99.99 | 99.96 | 100 | 99.98 | 100 | 99.98 | 100 | 100 | 99.99 | 99.94 |
| 24 | 100 | 99.99 | 100 | 100 | 99.99 | 99.97 | - | 100 | - | 99.99 | - | - | 100 | 99.96 |
| 25 | - | 99.99 | - | - | 100 | 99.99 | - | - | - | 99.99 | - | - | - | 99.98 |
| 26 | - | 100 | - | - | - | 100 | - | - | - | 99.99 | - | - | - | 99.99 |
| 27 | - | - | - | - | - | - | - | - | - | 100 | - | - | - | 99.99 |
| 28 | - | - | - | - | - | - | - | - | - | - | - | - | - | 100 |
| Avg | 8.72 | 9 | 9.53 | 10.31 | 10.87 | 11.24 | 10.07 | 10.95 | 11.73 | 12.46 | 10.3 | 11.14 | 12.02 | 12.8 |
| $E(n)$ | 9.08 | 9.41 | 9.87 | 10.62 | 11.24 | 11.57 | 10.36 | 11.2 | 12 | 12.75 | 10.54 | 11.41 | 12.26 | 13.06 |

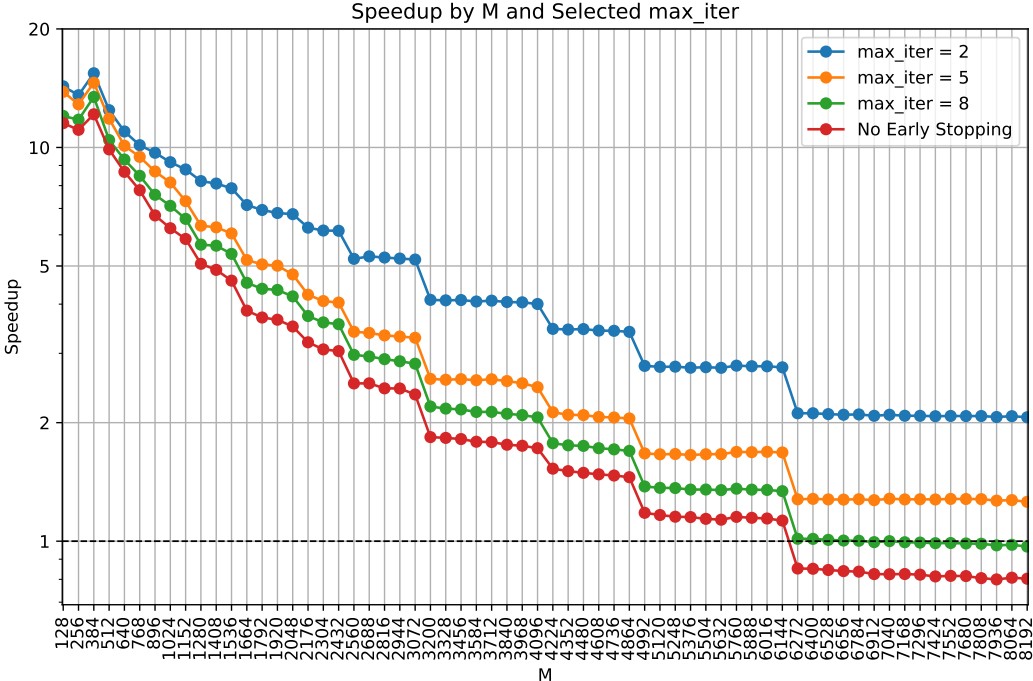

Figure 6: Speedup of RTop-K relative to PyTorch for different vector sizes $M$ and number of vectors $N = 65536$. The average speedup for each case is computed over $k = 64, 128, 256, 512$ and $k < M$. Precision $\epsilon = 10^{-16}$ is used for the no early stopping version.

$$E(n) = \log_2\left(2M\sqrt{\frac{\ln M}{\pi}}\right) - \frac{1}{2\ln 2}\left(\Phi^{-1}\left(1 - \frac{k}{M}\right)\right)^2 < \log_2\left(2M\sqrt{\frac{\ln M}{\pi}}\right) = O(\log M)$$

In each iteration, a reduction operation of length $M$ is required. Since one warp is fixed for one vector, the time complexity of this reduction is equivalent to serial reduction, which is $O(M)$. Therefore, the total time complexity of Algorithm 1 is $O(M \log M)$. In contrast, PyTorch's underlying operation, RadixSelect, has a time complexity of $O(M)$. Thus, when $M$ is sufficiently large, Algorithm 1 will lag behind traditional algorithms.

However, from a practical perspective, for $M \leq 8192$, as shown in Table 5, the growth of $E(n)$ is very slow. Additionally, since the search range has a lower bound (depending on the data type), the number of search iterations has an upper bound. Moreover, the searching stage is fully executed in shared memory, which leads to a decreasing proportion of time spent in the searching stage compared to the loading and selecting stages, as these involve increasing global memory accesses with the growth of $M$.

Therefore, we believe the actual time complexity of Algorithm 1 is less than $O(M \log M)$. As $M$ increases, the relative speedup of RTop-K compared to PyTorch decreases primarily because the relative efficiency of PyTorch's RadixSelect improves (the proportion of its initialization, histogram construction, and indexing overhead decreases).

We also evaluated the performance of Algorithm 1 with different precision settings, and the results are shown in Figure 7.

We found that precision has almost no impact on speed. Even with the setting of $\epsilon = 0$, as shown in Table 5, Algorithm 1 exits within 16 iterations in the vast majority of cases, and the remaining rare cases have a negligible impact on the overall performance. This is also because the searching stage is fully executed in shared memory, making Algorithm 1 less sensitive to the number of search iterations.

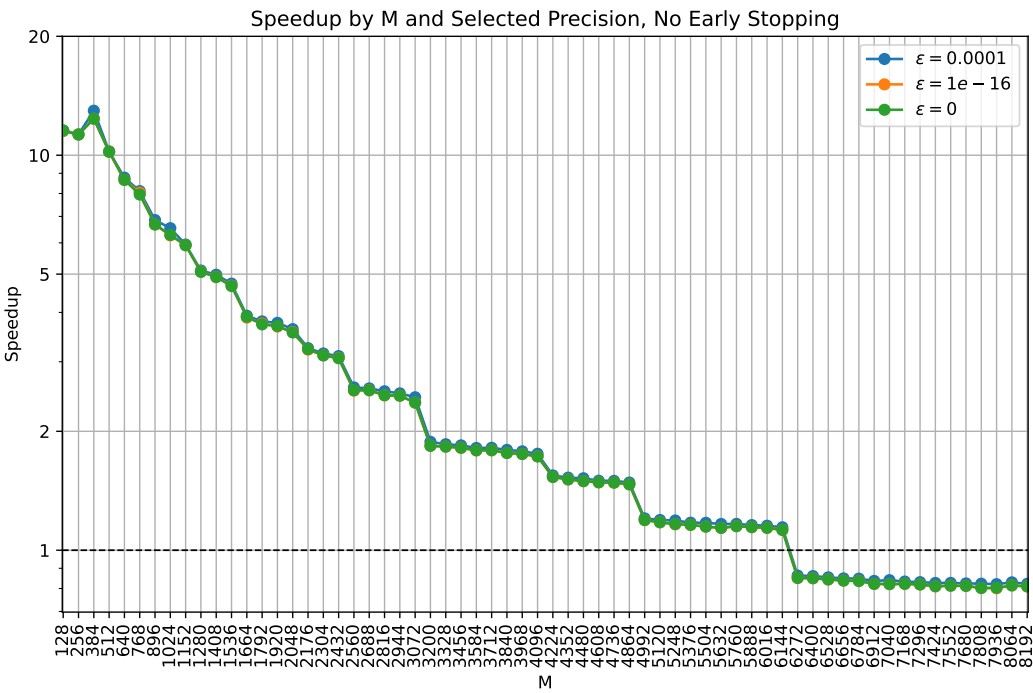

Figure 7: Speedup of RTop-K (no early stopping version) relative to PyTorch for different vector sizes $M$ and different precisions, with the number of vectors $N = 65536$. The average speedup for each case is computed over $k = 64, 128, 256, 512$ and $k < M$.

