# OpenReview forum: "RTop-K: Ultra-Fast Row-Wise Top-K Selection for Neural Network Acceleration on GPUs"
_ICLR.cc/2025/Conference — ICLR 2025 Poster_

### Official Review · Reviewer_TSWj · 2024-11-02

**Soundness:** 2
**Presentation:** 2
**Contribution:** 1
**Rating:** 5
**Confidence:** 3

**Summary:**

This paper presents RTop-K, a GPU-based algorithm that enhances the efficiency of row-wise top-K selection, specifically targeting Graph Neural Network (GNN) applications. The authors use a binary search strategy with optional early stopping to streamline top-K selection, offering a significant reduction in computational demands while preserving accuracy. Experiments show RTop-K provides speedups of 4.25× to 9.51× relative to PyTorch’s top-K implementation and achieves up to 31.53% faster training when integrated into GNN workflows.

**Strengths:**

1,Relevance and Novelty: The paper addresses an important computational bottleneck in GNN acceleration, offering an innovative approach with practical speed improvements for real-world applications.

2,Comprehensive Experimental Validation: The experimental analysis is thorough, showing RTop-K’s efficacy both as a standalone kernel and within the GNN training workflow across various architectures and datasets.

3,Effective Use of Early Stopping: The early stopping mechanism is a valuable addition, providing computational savings without significantly impacting model accuracy.

4,Reproducibility: The paper provides detailed descriptions of the algorithm and implementation, along with pseudocode, which improves replicability and clarity for the audience.

**Weaknesses:**

1,Ablation Study on Implementation Optimization: The paper describes a GPU implementation with several stages (loading, searching, and selecting) but could benefit from a breakdown of how each stage contributes to the overall speedup. A detailed analysis of performance gains per stage would shed light on optimization details. Additionally, an ablation study examining how binary search and early stopping impact efficiency would highlight their specific contributions and help readers better understand the trade-offs involved.

2,Scalability Results on Hidden Dimensions: While the paper provides scalability results for different values of K, it lacks scalability analysis on various hidden dimensions, which is equally important for understanding the algorithm’s performance on high-dimensional data, as seen in some GNN and non-GNN applications.

3,Guidance on Hyper-Parameter Tuning: The RTop-K algorithm involves critical hyper-parameters, specifically the precision value (ϵ') in Algorithm 1 and max_iter in Algorithm 2. Some discussion on how to set these parameters effectively across different tasks would enhance usability. A sensitivity analysis for these parameters could guide users in balancing performance with computational efficiency.

4,Lack of Comprehensive Baselines: The paper compares RTop-K primarily with PyTorch’s row-wise top-K implementation. Including comparisons with other top-K selection algorithms, such as Avg-TopK and TopK-SGD, which are popular in neural network training, would provide a more complete evaluation of RTop-K’s performance and position it within the broader field.

5,Discussion on Broader Applicability: While the paper focuses on GNN applications, top-K selection is widely used in various domains, such as expert selection in Mixture of Experts and vector search in large models like GPTs. A discussion on potential extensions of RTop-K to other contexts would strengthen the paper, showcasing its broader utility across different neural network architectures and tasks.

6.Contribution. summary of something, in this case, GPU top-K selection alrorithms, should not be the contribution of one research paper.

**Questions:**

Except for the weaknesses above, I have several questions about this work.

1, Just integrating the binary search into the top-K selection sounds not innovative, so can the authors dig more novelty from this work? Besides, I am curious that what searching performance improvement will using binary search bring, compared with not using binary searching.

2, The authors claim the parallel top-K selection algorithm, but what are the specific 'parallel' mechanism is NOT clear. It seems that, just many single top-K selection are executed on GPU platform. Can the authors explain this? If my understanding is correct, then this paper should not emphasize the "parallel"?

---

> ### Author Response · Authors · 2024-11-29
>
> Thank you for taking the time to read and review our paper!
>
> > Q1 - A breakdown of each stage
>
> We use the `clock64()` method within the kernel to roughly measure the clock cycles occupied by each stage and calculate their relative proportions.
>
> | Max_iter |    2   |    3   |    4   |    5   |    6   |    7   |    8   |
> |:---------|-------:|-------:|-------:|-------:|-------:|-------:|-------:|
> | Stage 1  | 40.99% | 34.78% | 31.62% | 30.38% | 29.50% | 28.47% | 27.66% |
> | Stage 2  | 29.64% | 36.75% | 41.20% | 45.19% | 48.62% | 51.46% | 53.57% |
> | Stage 3  | 29.37% | 28.47% | 27.18% | 24.42% | 21.88% | 20.07% | 18.77% |
>
> It can be observed that as `max_iter` increases, the time spent on the loading and selecting stages remains relatively constant, while the time spent on the searching stage continues to grow. This aligns with our design expectations. However, it is important to note that measuring clock cycles within the kernel using `clock64()` is limited to individual threads, and the results represent an average across all threads. Additionally, the total clock cycles measured within the kernel translate to a runtime that is shorter than the standard timing test we use for overall kernel execution. This is because the standard timing test (used for comparison with PyTorch) includes kernel on/off-chip overhead and some synchronization costs.
> The implementation and optimization effectiveness of the kernel depend on our skillful use and deep understanding of warp-level primitives, as well as efficient management of threads, warps, blocks, registers, and shared memory. In this context, the measurement of clock cycles within the kernel serves primarily as a validation of the optimization effectiveness.
>
>
> > Q2 - Scalability results on hidden dimensions
>
> In the newly added **Appendix A**, we successfully derived the expected number of iterations for RTopK's no early stopping version (Algorithm 1):
>
> $$
> E(n) = \log_2 \left( 2M \sqrt{\frac{\ln M}{\pi}} \right) - \frac{1}{2 \ln 2} \left( \Phi^{-1}\left( 1 - \frac{k}{M} \right) \right)^2
> $$
>
> According to our analysis, the time complexity of Algorithm 1 for single-vector top-k selection is less than $O(M\log M)$ and slightly greater than $O(M)$. This establishes the relationship between the time complexity and the hidden dimension $M$. For the overall row-wise top-k operation, which includes $N$ rows, the total time complexity ranges from $O(NM)$ to $O(NM\log M)$.
>
> In the newly added **Appendix B**, we performed an in-depth analysis of RTopK's performance under different values of $M$. For the no early stopping version:
> - When $M$ is below 3072, the speed of RTop-K is generally more than twice that of PyTorch.
> - When $M$ is between 3072 and 6144, RTop-K is still generally faster than PyTorch.
> - For $M > 8192$, our current shared memory-based acceleration strategy cannot be directly applied on the A6000 GPU due to shared memory limitations.
>
> From these findings, we conclude that RTopK holds a clear advantage in common neural network scenarios.
>
>
> > Q3 - Discussion on precision and max_iter
>
> First, please allow us to share our intriguing discovery: in the no early stopping version, the choice of precision has almost no impact on speed, even when set to maximum precision for floating-point type ($\epsilon = 0$). When $\epsilon = 0$, RTopK performs as an accurate top-k selection and is still several times faster than PyTorch in common neural network scenarios. For more details, please refer to the latter part of **Appendix B**.
>
> Therefore, the optimal precision choice for Algorithm 1 is simply $\epsilon = 0$.
>
> For Algorithm 2 (early stopping version), it can be observed from Figure 5 that early stopping has virtually no impact on accuracy; in some cases, early stopping even achieves higher accuracy than no early stopping (possibly due to the additional non-linearity introduced by early stopping). Based on the experimental results, setting `max_iter` to 4 or 5 strikes a good balance between acceleration and accuracy. A more detailed analysis would depend on additional practical applications of the row-wise Top-K layer.

---

> ### Author Response · Authors · 2024-11-29
>
> > Q4 - Compare with other top-K solutions
>
> First, it is worth noting that Avg-TopK and TopK-SGD are applications of top-k in neural networks rather than GPU implementations.
>
> Next, we want to make it clear that PyTorch's row-wise top-K implementation represents the state-of-the-art solution for the row-wise top-K scenario. Currently, other top-K solutions, including the reference [1] (Shanbhag et al.) provided by **Reviewer 3azA**, as well as those mentioned in our paper—Dr. Top-k (Gaihre et al.), AIR Top-k (Zhang et al.), and RadiK (Li et al.)—cannot be directly applied to row-wise top-K scenarios. If they must be applied, for example, in a row-wise top-K scenario with 65,536 rows, the kernel of the above solutions would need to be called 65,536 times to process the rows. The benchmark results are as follows (number of rows $N=65536$, vector size $M=[1024,2048]$):
>
> |  k  |    M | Refer-RadixSelect (ms)   | Refer-Bitonic (ms)   |   PyTorch-Topk (ms) |   RTopK (ms) |
> |:---:|:-----:|:-------------------------:|:---------------------:|:--------------------:|:-------------:|
> | 64  | 1024 | 8532.79                  | 1539.44              |                4.15 |         2.04 |
> | 64  | 2048 | 8644.20                  | 1572.86              |                6.09 |         4.46 |
> | 128 | 1024 | Error                    | Error                |                4.10 |         2.06 |
> | 128 | 2048 | 8591.77                  | 1606.94              |                6.09 |         4.53 |
> | 256 | 1024 | Error                    | Error                |                5.85 |         2.04 |
> | 256 | 2048 | Error                    | Error                |                7.83 |         4.48 |
> | 512 | 1024 | Error                    | Error                |                6.06 |         2.04 |
> | 512 | 2048 | Error                    | Error                |                8.01 |         4.47 |
>
> The reference [1] (Shanbhag et al.) includes both RadixSelect and Bitonic kernels. It supports a minimum vector size $M = 1024$, but errors occur when the ratio of $k/M$ reaches a certain threshold.
>
> As shown, the reference implementation is hundreds of times slower than both the PyTorch implementation and our implementation. This demonstrates that top-K implementations designed for single large vectors (typically with vector sizes in the millions) are not directly applicable to row-wise top-K scenarios (large batches of small vectors, which are common in deep learning and GNNs). Our innovations and contributions lie in designing and optimizing specifically for the row-wise top-K scenario.
>
> [1] Anil Shanbhag et al. Efficient Top-K Query Processing on Massively Parallel Hardware.
>
>
> > Q5 - Discussion on broader applicability
>
> Thanks for the suggestions. The RTop-K can indeed be used for Kth value search in Mixture of Expert (MoE) router network computation and vector search in the Retrieval Augmented Generation (RAG) application, especially for workloads with many short vectors to process. In the large batch training stage of LLMs [1] or the diffusion transformer models [2], with a batch size of 200 and a sequence length of 1000, there could easily be 0.2 million tokens to process for each Mixture of Experts layer. In such cases, the row-wise TopK operation could potentially become a bottleneck. We will add a section discussing these broader applications in the final version of the paper.
>
> [1] Dai, D., Deng, C., Zhao, C., Xu, R. X., Gao, H., Chen, D., ... & Liang, W. (2024). Deepseekmoe: Towards ultimate expert specialization in mixture-of-experts language models. arXiv preprint arXiv:2401.06066.
>
> [2] Sun, H., Lei, T., Zhang, B., Li, Y., Huang, H., Pang, R., ... & Du, N. (2024). EC-DIT: Scaling Diffusion Transformers with Adaptive Expert-Choice Routing. arXiv preprint arXiv:2410.02098.
>
>
> > Q6 - Contribution
>
> Our contribution lies in the following key aspects:
> 1. Optimizing the row-wise TopK kernel to achieve performance several times faster than PyTorch.
> 2. Integrating RTopK into GNNs training systems, resulting in up to a 1.3x overall speedup.
>
> Optimizing the row-wise TopK kernel itself is a non-trivial research problem, as row-wise TopK is challenging to accelerate and can become a bottleneck in GNN training (as shown in Table 4, where it accounts for 12%-27% of total training time).
>
> We respectfully disagree with the notion that *GPU top-K selection algorithms themselves should not be the contribution of a research paper*. GPU top-K selection is a highly impactful research area that supports numerous applications. In fact, there are already multiple research papers [1][2][3] dedicated to optimizing GPU top-K selection algorithms.
>
> [1] Anil Gaihre et al. Dr. Top-k: Delegate-centric Top-k on GPUs. SC '21.
> [2] Jingrong Zhang et al. Parallel top-k algorithms on GPU: A comprehensive study and new methods. SC ’23.
> [3] Yifei Li et al. RadiK: Scalable and optimized GPU-parallel radix top-k selection. ICS ’24.

---

> ### Author Response · Authors · 2024-11-29
>
> > Q7 - Advantages of binary search algorithm
>
> The advantages of binary search-based top-k selection lie in its simplicity and resource efficiency, making it more effective in row-wise top-k scenarios and even several times faster than PyTorch's official implementation.
>
> Here, we compare with two traditional algorithms, RadixSelect and Bitonic, as presented in the reference [1] (Shanbhag et al.) provided by **Reviewer 3azA**:
>
> - **RadixSelect**
>   Refer to the implementation: [radixSelectTopK.cuh](https://github.com/anilshanbhag/gpu-topk/blob/master/src/radixSelectTopK.cuh).
>   RadixSelect involves a series of steps, including initialization, histogram construction, indexing, sorting and selection, and extraction of the top-K elements.
>
> - **Bitonic**
>   Refer to the implementation: [bitonicTopK.cuh](https://github.com/anilshanbhag/gpu-topk/blob/master/src/bitonicTopK.cuh).
>   Bitonic sorting requires multiple rounds of extensive element exchanges and includes memory read-write dependencies.
>
> In contrast, binary search-based top-k selection, during its main stage (the searching stage), involves only parallel counting operations. These operations can efficiently utilize warp-level primitives for reduction, with no element movement or memory read-write dependencies. This makes binary search-based top-k selection significantly faster for small vectors compared to traditional algorithms.
>
> However, due to time complexity considerations, as the vector size grows beyond a certain point, RTopK may lag behind traditional algorithms. Nonetheless, RTopK retains its advantage in common neural network scenarios.
>
> [1] Anil Shanbhag, Holger Pirk, and Samuel Madden. Efficient Top-K Query Processing on Massively Parallel Hardware.
>
>
> > Q8 - About "parallel"
>
> The parallelism of the RTopK implementation for the row-wise top-K scenario is reflected on two levels:
>
> 1. **Intra-warp parallelism**:
>    RTopK uses a single warp to process one row (a limited vector). During this process, the loading, searching, and selecting stages involve collaboration among all threads within the warp. Specifically, operations such as tree-reduction and Ballot rely on warp-level primitives to achieve efficient parallel processing within the warp.
>
> 2. **Inter-row parallelism**:
>    The RTopK kernel processes all rows of the matrix in parallel. Since the computation and memory resources allocated to each row's processing are mutually independent, the parallelism at this level is relatively straightforward. However, we still need to carefully allocate resources to each block to maximize the efficiency of inter-row parallelism. As mentioned in Appendix B, we allocate ⌊8192/M⌋ warps per block, which has proven to be an effective strategy.
>
> While our primary focus is on designing and optimizing the algorithms, parallelism is also an important aspect of our design.

---

> ### Author Response · Authors · 2024-12-01
> **Gentle reminder**
>
> Dear Reviewer,
>
> Thank you once again for your valuable feedback, we have carefully addressed your concerns and incorporated your suggestions. In the rebuttal, we added many other ablation and comparative studies ensuring that RTopK is thoroughly studied. Furthermore, we have open-sourced our work (currently in anonymous github).
>
> As the discussion phase is nearing its end, we would like to kindly follow up to confirm whether our response has sufficiently addressed your concerns. If so, we would greatly appreciate it if you could consider raising the score further to *6: marginally above the acceptance threshold*. We firmly believe that our work has the potential to contribute significantly to the field and meets the criteria for acceptance. If fortunate enough to receive the reviewers' recognition, we are committed to contributing this work to the official **PyTorch library** to benefit the broader community.
>
> Please do not hesitate to let us know if you have any additional questions or concerns; we would be happy to address them.
>
> Thank you again for your time and thoughtful input.

---

### Official Review · Reviewer_ZVep · 2024-11-02

**Soundness:** 3
**Presentation:** 3
**Contribution:** 3
**Rating:** 6
**Confidence:** 2

**Summary:**

RTop-K is an efficient parallel row-wise top-k selection algorithm for GPUs, leveraging a binary search-based approach to significantly enhance performance in neural network training and inference. This algorithm achieves speedups ranging from 4.25 to 9.51 times compared to existing GPU top-k implementations, and it also improves the overall training workflow of MaxK-GNN by 9.76% to 31.53%. Furthermore, RTop-K, when combined with the Early Stopping technique, maintains model accuracy while enhancing efficiency through reduced resource requirements.

**Strengths:**

1. RTop-K shows a 4.25- to 9.51-fold speedup over existing state-of-the-art GPU-based row-wise top-k implementations, demonstrating in particular performance improvements in GNN training and inference.  It is meaningful that RTop-K is designed based on efficient parallel processing and resource optimization.
2. RTop-K adopts an approximation method to select the top k elements from each row without sorting, reducing computational time complexity and accelerating neural network training.

**Weaknesses:**

1. What is the method to overcome the limitations of approximation methods used by algorithms?
2. Is resource optimization a general methodology? In the paper, the performance of RTop-K is likely to depend on a particular GPU architecture (A6000). Therefore, it is necessary to review whether usabilty is secured in other GPU environments or structures.
3. RTop-K is optimized for limited-sized vectors, but there seems to be a lack of discussion on how resource consumption and memory management are performed on very large datasets.

**Questions:**

See weakness.

---

> ### Author Response · Authors · 2024-11-29
>
> Thank you for taking the time to read and review our paper!
>
> > Q1 - Overcome the limitations of approximation
>
> First, please allow us to share our intriguing discovery: in the no early stopping version, the choice of precision has almost no impact on speed, even when set to maximum precision for floating-point type ($\epsilon = 0$). When $\epsilon = 0$, RTopK performs as an accurate top-k selection and is still several times faster than PyTorch in common neural network scenarios.
>
> For more details, please refer to the newly added **Appendix A and B**. This significantly boosts our confidence in contributing our work to PyTorch's official library, though it will still demand considerable engineering efforts.
>
> To completely overcome the limitations of approximation, using the accurate version provides the optimal experience!
>
> Of course, the early stopping version is also a viable choice. In Table 2, we present theoretical simulation tests for the relative error and hit rate of the approximation algorithm. The analysis shows that the relative error of the approximation algorithm can easily be controlled within 5%. Tests on the impact of this approach on actual test accuracy are shown in Figure 5 (blue dots represent the accuracy with early stopping, and the green horizontal line represents the accuracy without early stopping). It can be observed that early stopping has almost no impact on accuracy. Therefore, when applying the row-wise top-k layer in typical models, early stopping can be used to achieve greater speedup with minimal impact on accuracy.
>
>
> > Q2 - Testing on other GPUs
>
> As shown in **Appendix B**, RTopK generally achieves speeds more than twice that of PyTorch when the vector size $M < 3072$. This is sufficient to cover the vast majority of row-wise top-k use cases in neural networks and is applicable to other mainstream GPUs.
> We extended Table 3 to include tests on A100 and Quadro RTX 6000 GPUs. The results are as follows:
>
> ### A100:
> | Max Iteration   |     2 |     3 |     4 |    5 |    6 |    7 |    8 |   No Early Stopping |
> |:----------------|------:|------:|------:|-----:|-----:|-----:|-----:|--------------------:|
> | $M=256$        | 15.80 | 12.84 | 10.83 | 9.45 | 8.39 | 7.66 | 7.19 |                6.58 |
> | $M=512$        |  8.43 |  6.56 |  5.38 | 4.59 | 4.01 | 3.71 | 3.40 |                2.93 |
> | $M=768$        |  9.45 |  7.47 |  6.19 | 5.30 | 4.66 | 4.18 | 3.82 |                3.20 |
> | **Average**    | 11.23 |  8.96 |  7.47 | 6.45 | 5.68 | 5.18 | 4.80 |                4.24 |
>
> ### Quadro RTX 6000:
> | Max Iteration   |     2 |    3 |    4 |    5 |    6 |    7 |    8 |   No Early Stopping |
> |:----------------|------:|-----:|-----:|-----:|-----:|-----:|-----:|--------------------:|
> | $M=256$        | 10.12 | 8.06 | 6.78 | 5.85 | 5.18 | 4.72 | 4.42 |                4.05 |
> | $M=512$        |  6.93 | 5.34 | 4.36 | 3.71 | 3.27 | 3.01 | 2.77 |                2.41 |
> | $M=768$        |  7.27 | 5.79 | 4.80 | 4.14 | 3.66 | 3.30 | 3.03 |                2.55 |
> | **Average**    |  8.11 | 6.40 | 5.31 | 4.57 | 4.04 | 3.68 | 3.41 |                3.00 |
>
> It can be observed that RTopK outperforms PyTorch on both A100 and Quadro RTX 6000 GPUs.
>
>
> > Q3 - Resource and memory analysis on large datasets
>
> RTopK is a fast operator for computing row-wise top-k. In its application scenarios, while the row-wise top-k operation has a certain computational complexity, it is typically not relatively costly in terms of resource and memory usage.
>
> For example, in MaxK-GNN, the SpMM operation is the primary consumer of memory, whereas the memory usage of RTopK is comparatively minimal. Consider the Reddit graph, which has 232,965 nodes and 114,615,891 edges. With a hidden dimension of 256, the total memory traffic for a single SpMM operation is 13.13 GB, while the total memory traffic for a single row-wise top-k operation is only 0.29 GB.
>
> Thus, our focus is primarily on optimizing the runtime of RTopK, as its memory resource requirements will not become a bottleneck.

---

> ### Author Response · Authors · 2024-12-02
>
> We would like to further clarify question 3.
> > Q3 - Resource and memory analysis on large datasets
>
> RTop-K is designed for the row-wise TopK operator over a large number of small vectors, which can already represent large datasets. For instance, the Ogbn-products dataset used in the experiment (Table 4/Figure 5 in the paper) contains $N = 2,449,029$ rows. With an embedding dimension of $M = 1024$, the embedding matrix (input matrix in the row-wise TopK scenario) comprises 2,507,805,696 elements, occupying 10.03 GB of memory. This already qualifies as a large dataset.
>
> Additionally, for the large number of rows $N$ with large vector size $M$, we have added profiling results up to $N = 65536$, $M \leq 8192$. For details, please see Figure 6 in Appendix B. With the early stop algorithm, RTop-K kernel consistently outperforms Pytorch kernel.
>
> For memory resource usage, we can classify the memory resource usage into two types: global memory allocation and global memory traffic (read/write).
>
> For global memory allocation, RTop-K kernel requires no additional global memory allocation. The memory usage of RTop-K  is even lower than that of conventional TopK Pytorch kernel, as it will only require necessary memory allocation for input matrix (batch of vectors) and output matrix (batch of vectors).
>
> In terms of global memory traffic (read/write) during the kernel computation, RTop-K consumes significantly less memory traffic compared to other operators in the neural network workflow. It involves only $N \times M$ global memory reads and $N \times k$ global memory writes, which represent a memory traffic similar to that of the element-wise operator, such as ReLU. All other operations are completed within shared memory.
>
> For example, in the Reddit graph, which has 232,965 nodes and 114,615,891 edges. With a hidden dimension of 256, the total global memory traffic (read/write) for a single row-wise top-k operation is only 0.29 GB (memory traffic, not additional static memory usage). While in MaxK-GNN, other operators such as SpMM, a single SpMM forward consumes totally 13.13 GB memory traffic(read/write).
>
> Thus, our focus is primarily on optimizing the runtime of RTop-K, as the kernel itself doesn’t require additional global memory allocation other than necessary input/output memory, and it requires very minimum global memory traffic usage, so the memory resource will not become a bottleneck.

---

> ### Author Response · Authors · 2024-12-02
> **Gentle reminder**
>
> Dear Reviewer,
>
> We hope our response has addressed all of your concerns. Please feel free to let us know if you have any additional questions or feedback.
> This project is already open-sourced (currently in anonymous github), and we assure you that we will do our utmost to contribute this work to the official **PyTorch library**. If you could kindly consider raising your evaluation, your support would greatly help accelerate this process. Thank you very much for your time and thoughtful consideration.

---

### Official Review · Reviewer_4z6F · 2024-11-02

**Soundness:** 3
**Presentation:** 4
**Contribution:** 4
**Rating:** 10
**Confidence:** 4

**Summary:**

This paper looks at performing top-k row-wise selection in matrices that are, broadly, tall and skinny – with fewer than 1000 elements per row but perhaps millions of rows. Most existing algorithms focus on the opposite regime – relatively few rows that are much longer – and this distinction is important for designing performant GPU implementations. The authors introduce an algorithm that seeks to outperform alternative approaches, including the default RadixSelect algorithm in PyTorch, and is designed for the hardware, using warp-level parallelism within a row and careful use of the cache hierarchy. They also introduce an early stopping version of the algorithm that uses a fixed number of iterations, as load imbalance across warps/rows could significantly decrease the potential speedup. Results are shown in two settings. First, they consider the performance of the algorithm with and without early stopping on synthetic data, evaluating the number of iterations required and finding acceptable error rates when using early stopping. When compared with PyTorch’s implementation, they find substantial speedups (2-6x). Second, the implement their top-k in a few max-k GNN algorithms, like SAGE, where top-k calculations account for a non-trivial amount of runtime, and test on a few benchmark graph datasets. They report speedups of 10-30% and, with early stopping, find that model test accuracy fluctuates around that expect from using the optimal top-k values.

**Strengths:**

This is a good study. Despite the field being well-studied, their approach is fairly original. It really sets aside what has typically been done and asks, for this context, precisely what does the algorithm need to do and on what hardware would it run. The consideration of the appropriate level of parallelism based on both GPU hardware constraints and problem size considerations is thoughtful, and they get to a nice answer. The paper is quite well written and was pleasantly easy to read.

The GNN test doesn’t have a ton of detail, but is well conducted and does a nice job of rounding out a clear ML story from an otherwise pure HPC-focused study.

**Weaknesses:**

I debate whether this is a better baseline comparison that PyTorch’s RadixSelect. As the authors suggest, it solves a somewhat more difficult problem – sorting. I don’t disagree with the comparison for the GNNs, as it is the operational default, but it was more difficult for me to interpret the value of the speedups is in the synthetic data top-k test.

I would have liked the authors to push the bounds on the algorithm a little more. I understand the regime they tested is representative of a standard family of networks, but I would have liked to see another version of at least Figure 4 which looked at increasing M to ~10^4 or even 10^5 . This would indicate potential places where GPU cache/warp size limitations cause behavior to deteriorate, or suggest where the line between better performance with this algorithm vs more traditional approaches like RadiK is crossed.

**Questions:**

The authors did a nice job of taking an HPC paper and providing a compelling presentation for an ML setting. But, as suggested above, I feel it would be a better paper in itself (if admittedly not necessarily for this venue) if the authors considered how it would perform for large M and more general guidance could be given to researchers on where to switch between algorithms.

---

> ### Author Response · Authors · 2024-11-29
>
> Thank you very much for your recognition of our work. Your suggestions are highly reasonable, as traditional algorithms like RadixSelect can outperform our method when the vector size $M$ grows beyond a certain point. Determining the critical switching point is indeed necessary.
>
> Please refer to the newly added **Appendix A**, where we successfully derive the expression for the expected number of iterations in Algorithm 1 (no early stopping version):
>
> $$
> E(n) = \log_2 \left( 2M \sqrt{\frac{\ln M}{\pi}} \right) - \frac{1}{2 \ln 2} \left( \Phi^{-1}\left( 1 - \frac{k}{M} \right) \right)^2
> $$
>
> According to our analysis, the time complexity of Algorithm 1 is less than $O(M\log M)$ and slightly greater than $O(M)$. In comparison, the underlying RadixSelect in PyTorch has a time complexity of $O(M)$. Of course, this is purely theoretical. The actual situation is more complex. Please refer to **Appendix B**. For the no early stopping version, when $M$ is below 3072, the speed of RTop-K is generally more than twice that of PyTorch. When $M$ is between 3072 and 6144, RTop-K is generally faster than PyTorch. (For $M > 8192$, our current shared memory-based acceleration strategy cannot be directly applied on the A6000 GPU due to shared memory limitations.)
>
> We are also delighted to share: the choice of precision has almost no impact on the speed of the no early stopping version, even when set to maximum precision for floating-point type ($\epsilon = 0$). When $\epsilon = 0$, RTopK performs as an accurate top-k selection and is still several times faster than PyTorch in common neural network scenarios. This significantly boosts our confidence in contributing our work to PyTorch's official library, though it will still demand considerable engineering efforts.
>
> We also notice that when $M$ is a multiple of 256 or 512, the performance of RTopK drops. This is likely due to shared memory bank conflicts, and we are confident in addressing this issue in the final version of the paper (possibly by adding padding in the data access structure to break the conflict pattern).
>
> Thank you again for recognizing our work! May we ask for your even stronger approval? :)

---

> > ### Comment · Reviewer_4z6F · 2024-12-02
> >
> > Thank you for the response. I think the additional analysis is very helpful and I think it clearly addresses the question of broader applicability that I and other reviewers raised.

---

> > > ### Author Response · Authors · 2024-12-03
> > >
> > > Dear Reviewer,
> > >
> > > We sincerely thank you for your full support of our work!
> > >
> > > We assure you that we will try our best to contribute this work to the official PyTorch library to benefit the community as a way of giving back for your support.
> > >
> > > Best regards,
> > >
> > > The Authors

---

### Official Review · Reviewer_3azA · 2024-11-03

**Soundness:** 3
**Presentation:** 3
**Contribution:** 2
**Rating:** 5
**Confidence:** 2

**Summary:**

This paper presents RTop-K, a GPU acceleration of row-wise TopK selection. The proposed algorithm is based on binary search. Also, it introduces the early stopping strategy to improve the performance further. The experimental results show that it outperforms the state of the art row-wise TopK implementation.

**Strengths:**

1. The paper is well organized and easy to follow.
2. This work addresses one of the key component in the GNN training.

**Weaknesses:**

1. Regarding the kernel performance comparison, the authors compares the solution with the PyTorch. The proposed solution should be compared with other topK solutions, like Shanbhag et al. [1], or radix select-based etc--assessment that "not suitable for GNN is not sufficient", in my opinion. Besides, I am not sure the recent PyTorch is the best baseline to compare, as I see some complaints that it is slower than torch.sort (I am not sure whether this is already resolved or not).

[1] Anil Shanbhag, Holger Pirk, and Samuel Madden. Efficient Top-K Query Processing on Massively Parallel Hardware.

2. Limited applicability: the experiment shows the speedup for Max K-GNN (includes GraphSAGE, GIN, GCN) in the only. Not sure if there's a broader impact on other DL networks.
3. Theoretical analysis on early stopping.
4. It'd be great if you can mark speedup on Fig. 4
5. How the implementation is close from the roofline? Can you please share the Nsight profiling results?
6. Is TopK bottleneck in the GNN? Can you please share the breakdown results, not just showing the portion of the TopK part?

**Questions:**

Please refer to Weakness above.

---

> ### Author Response · Authors · 2024-11-29
>
> Thank you for taking the time to read and review our paper!
>
> > Q1 - Compare with other top-K solutions
>
> We want to make it clear that PyTorch's row-wise top-K implementation represents the state-of-the-art solution for the row-wise top-K scenario. Currently, other top-K solutions, including the reference you provided (Shanbhag et al.), as well as those mentioned in our paper—Dr. Top-k (Gaihre et al.), AIR Top-k (Zhang et al.), and RadiK (Li et al.)—cannot be directly applied to row-wise top-K scenarios. If they must be applied, for example, in a row-wise top-K scenario with 65,536 rows, the reference kernel would need to be called 65,536 times to process the rows. The benchmark results are as follows (number of rows $N=65536$, vector size $M=[1024,2048]$):
>
> |  k  |   M   | Refer-RadixSelect (ms) | Refer-Bitonic (ms) | PyTorch-Sort (ms) | PyTorch-Topk (ms) | RTopK(no early stopping) (ms) |
> |:---:|:-----:|:-----------------------:|:------------------:|:-----------------:|:-----------------:|:----------:|
> |  64 |  1024 |        8532.79         |       1539.44      |       5.92        |       4.15        |     2.04   |
> |  64 |  2048 |        8644.20         |       1572.86      |      11.93        |       6.09        |     4.46   |
> | 128 |  1024 |         Error           |        Error       |       5.92        |       4.10        |     2.06   |
> | 128 |  2048 |        8591.77         |       1606.94      |      11.93        |       6.09        |     4.53   |
> | 256 |  1024 |         Error           |        Error       |       5.92        |       5.85        |     2.04   |
> | 256 |  2048 |         Error           |        Error       |      11.93        |       7.83        |     4.48   |
> | 512 |  1024 |         Error           |        Error       |       5.92        |       6.06        |     2.04   |
> | 512 |  2048 |         Error           |        Error       |      11.93        |       8.01        |     4.47   |
>
>
> The reference (Shanbhag et al.) includes both RadixSelect and Bitonic kernels. It supports a minimum vector size $M = 1024$, but errors occur when the ratio of $k/M$ reaches a certain threshold.
>
> As shown, the reference implementation is hundreds of times slower than both the PyTorch implementation and our implementation. This demonstrates that top-K implementations designed for single large vectors (typically with vector sizes in the millions) are not directly applicable to row-wise top-K scenarios (large batches of small vectors, which are common in deep learning and GNNs). Our innovations and contributions lie in designing and optimizing specifically for the row-wise top-K scenario.
>
> In the table, we also compare PyTorch-Sort and PyTorch-Topk. Result shows that PyTorch-Topk is faster than PyTorch-Sort in most cases, and is only slightly slower than PyTorch-Sort in only 1 case when $k/M$ is too large (k=512 M=1024). In both of those cases, our method RTop-K shows a significant advantage over both PyTorch-Sort and PyTorch-Topk.
>
>
>
> > Q2 - Applicability
>
> First, RTopK indeed has more application scenarios. For example, it can be used for Kth value search in Mixture of Expert (MoE) router network computation and vector search in the Retrieval Augmented Generation (RAG) application, especially for workloads with many short vectors to process. In the large batch training stage of LLMs [1] or the diffusion transformer models [2], with a batch size of 200 and a sequence length of 1000, there could easily be 0.2 million tokens to process for each Mixture of Experts layer. In such cases, the row-wise TopK operation could potentially become a bottleneck.
>
> Second, the row-wise TopK layer can generally be used as an non-linear activation layer in neural network models (currently used in our GNNs), similar to ReLU nonlinearity. Compared to ReLU, it can output a row-balanced structural sparsity pattern, which enables efficient utilization of this sparsity in applications like MaxK-GNN — a significant and successful example. On the other hand, it can provide a novel non-linearity, potentially leading to better theoretical accuracy for DNNs in certain scenarios.
>
> Thank you for your insight! We will add a section discussing these applications in the final version of the paper.
>
> [1] Dai, D., Deng, C., Zhao, C., Xu, R. X., Gao, H., Chen, D., ... & Liang, W. (2024). Deepseekmoe: Towards ultimate expert specialization in mixture-of-experts language models. arXiv preprint arXiv:2401.06066.
> [2] Sun, H., Lei, T., Zhang, B., Li, Y., Huang, H., Pang, R., ... & Du, N. (2024). EC-DIT: Scaling Diffusion Transformers with Adaptive Expert-Choice Routing. arXiv preprint arXiv:2410.02098.

---

> ### Author Response · Authors · 2024-11-29
>
> > Q3 - Theoretical analysis
>
> Thank you for your suggestion to include additional theoretical derivations!
>
> For Algorithm 1 (no early stopping version), we conducted mathematical derivations and more detailed theoretical simulations. We successfully derived an expression for the expected number of iterations:
>
> $$
> E(n) = \log_2 \left( 2M \sqrt{\frac{\ln M}{\pi}} \right) - \frac{1}{2 \ln 2} \left( \Phi^{-1}\left( 1 - \frac{k}{M} \right) \right)^2
> $$
>
> This expression matches the simulation results well, as detailed in the newly added **Appendix A**.
>
> One surprising finding is that setting the precision of the no early stopping version to maximum for floating-point type ($\epsilon=0$) has almost no impact on speed. When $M < 3072$, the speed of RTop-K ($\epsilon=0$) is generally more than twice that of PyTorch. Detailed results can be found in the newly added **Appendix B**. This significantly boosts our confidence in contributing our work to PyTorch's official library, though it will still demand considerable engineering efforts.
>
> For Algorithm 2 (early stopping version), since the specific execution process involves a series of discrete decisions, a more in-depth theoretical analysis was supported through simulation experiments. In **Table 2**, we conducted simulation tests on the relative error and hit rate of early stopping. The analysis shows that the relative error of early stopping can be easily controlled within 5%. **Figure 5** illustrates the impact of early stopping on the actual test accuracy of the model (blue dots represent the accuracy with early stopping, while the green horizontal line represents the accuracy without early stopping). The conclusion is that early stopping has almost no impact on accuracy; in some cases, the accuracy with early stopping is even higher than without it. Therefore, early stopping is well-suited for further accelerating the application of this algorithm in neural networks.
>
>
> > Q4 - Mark speedup on Fig. 4
>
> Got it, we will highlight the acceleration effect in Figure 4 of the final version of the paper.
>
> > Q5 - Nsight profiling results
>
> Below are the NSight Compute profiling results of RTopK (no early stopping, $\epsilon=0$) and PyTorch TopK, tested on an A6000 GPU with $N=65536$, $M=768$, and $k=64$:
>
> | Implementation                |   Compute (SM) Throughput (%) |   Memory Throughput (%) |   L1/TEX Cache Throughput (%) |   L2 Cache Throughput (%) |   DRAM Throughput (%) |
> |:------------------------------|------------------------------:|------------------------:|------------------------------:|--------------------------:|----------------------:|
> | RTopK                         |                         88.78 |                   88.78 |                         94.08 |                     26.3  |                 52.8  |
> | PyTorch                       |                              |                          |                               |                           |                       |
> | radixFindKthValues            |                         61.37 |                   67.32 |                         41.01 |                     26.09 |                 67.32 |
> | computeBlockwiseWithinKCounts |                         30.93 |                   17.6  |                         18.47 |                      2.44 |                  5.43 |
> | gatherTopK                    |                         60.32 |                   57.58 |                         58.62 |                     19.6  |                 44.96 |
>
> As shown, the RTopK kernel achieves very high Compute Throughput and Memory Throughput, with both being nearly balanced, fully leveraging the GPU's performance.
>
> In contrast, PyTorch's TopK implementation is split into multiple kernels, primarily consisting of several calls to `radixFindKthValues` and `computeBlockwiseWithinKCounts`, followed by a single `gatherTopK` call. None of these kernels fully utilize Compute Throughput and Memory Throughput.

---

> ### Author Response · Authors · 2024-11-29
>
> > Q6 - GNN & RTop-K training time breakdown
>
> Below is the breakdown of training time for GCN & RTop-K models on several graph datasets (with top-k acceleration, hidden dimension $M = 256$, $k = 32$):
>
> | Dataset           |       Yelp (#Nodes 716847, #Edges 13954819)        |          |        Reddit (#Nodes 232965, #Edges 114615891)        |          |        Ogbn-products (#Nodes 2449029, #Edges 123718280)        |          |       Flickr (#Nodes 89250, #Edges 989006)        |          |
> |--------------------|:-------------------------------------------------:|:--------:|:------------------------------------------------------:|:--------:|:----------------------------------------------------------:|:--------:|:-------------------------------------------------:|:--------:|
> |                    |                     PyTorch                      |   RTopK  |                        PyTorch                        |   RTopK  |                           PyTorch                          |   RTopK  |                     PyTorch                      |   RTopK  |
> | SpMM (accelerated) |                     16.4%                        |   21.1%  |                        55.1%                         |   61.3%  |                           33.9%                           |   40.8%  |                     10.9%                        |   14.2%  |
> | row-wise topk      |                     25.8%                        |    **4.5%**  |                        11.6%                         |    **1.8%**  |                           19.6%                           |    **3.2%**  |                     26.8%                        |    **4.9%**  |
> | other operators    |                     57.8%                        |   74.4%  |                        33.2%                         |   36.9%  |                           46.5%                           |    56%   |                     62.3%                        |   80.9%  |
> | total speedup      |                                             |     1.29x     |                                                |      1.11x     |                                                      |     1.20x     |                                            |     1.30x      |
>
> ### Is TopK a bottleneck in the GNN?
>
> The row-wise TopK operation serves as a nonlinear component in our GNN framework, introducing structural sparsity and improving performance. However, without a specialized design, the row-wise TopK itself can become a bottleneck during GNN training and inference. To mitigate this, we optimize the row-wise TopK kernel using the RTop-K implementation, achieving further acceleration.

---

### Author Response · Authors · 2024-11-29
**Global Response:**

We sincerely appreciate the reviewers’ valuable and constructive feedback. During the rebuttal phase, we have made a substantial effort to address the concerns by providing detailed mathematical derivations and enriching our experimental results. Additional contents are elaborated in **Appendix A and B** of the revised paper. Below are the highlights of our updated work:

1. **Derivation of the expected number of iterations for Algorithm 1 (no early stopping version):**
   $$
   E(n) = \log_2 \left( 2M \sqrt{\frac{\ln M}{\pi}} \right) - \frac{1}{2 \ln 2} \left( \Phi^{-1}\left( 1 - \frac{k}{M} \right) \right)^2
   $$
   The derived result matches well with experiment results.

2. **Detailed profiling of RTopK performance across a wider range of vector sizes $M$:**
   - The no early stopping version is generally more than 2 times faster than PyTorch when $M < 3072$.
   - It generally remains faster than PyTorch when $M < 6144$.

3. **Precision choice in Algorithm 1:**
   We discovered that the choice of precision has almost no impact on speed. Therefore, the precision can be set to maximum precision for floating-point type ($\epsilon = 0$), allowing RTopK to function as an accurate row-wise top-k selection operator, while still remaining several times faster than PyTorch in common scenarios.

---

### Meta-Review · Area_Chair_bGu7 · 2024-12-17

**Metareview:**

This paper presents a parallel row-wise top-k selection algorithm designed for GPUs  and demonstrate how it can accelerate the training workflow of graph neural networks (GNns).

In terms of strengths, reviewers appreciated that this work addressed a key bottleneck in GNN training. The experimental results were lauded. Reviewers felt the paper was well-written with one  reviewer stating that this work "does a nice job of rounding out a clear ML story from an otherwise pure HPC-focused study."

Most of the weaknesses outlined by reviewers were questions which the authors did a great job of answering, and the paper has been strengthened as a result of these discussions. Having read all the authors responses, I don't see any real outstanding weaknesses.

3/4 of the final review scores are borderline with 1 very strong accept (10,6,5,5). I notice however that the author's practically address all of Reviewer 3azA's concerns (and the reviewer moved from a 3->5). Despite the borderline reject I don't see any grounds for rejection in their review. Reviewer TSWj scored 5 but did not engage with the authors (or this AC) despite the extensive response and experiments that the authors provided. Because of this, I don't see either of these 5 reviews (both of which are low confidence) as a reason to not accept this paper when there are no real identifiable weaknesses, and the contributions are very much appreciated by the other reviewers.

**Additional Comments On Reviewer Discussion:**

Most concerns were questions which were well-addressed by the authors. Reviewer 3azA raised their score from a 3 to 5, but did not go up to a 6 when asked by the authors. Reviewer 4z6F changed their score from an 8 to a 10 after the authors responded to their concerns. The other two reviewers did not respond to the authors. Given that the author responses were so comprehensive and then went unacknowledged, I am placing less weight on these reviews as it looks like the matters were addressed (indeed, both are low confidence too).

---

### Decision · Program_Chairs · 2025-01-22

Accept (Poster)